# PROXY-GS: EFFICIENT 3D GAUSSIAN SPLATTING VIA PROXY MESH

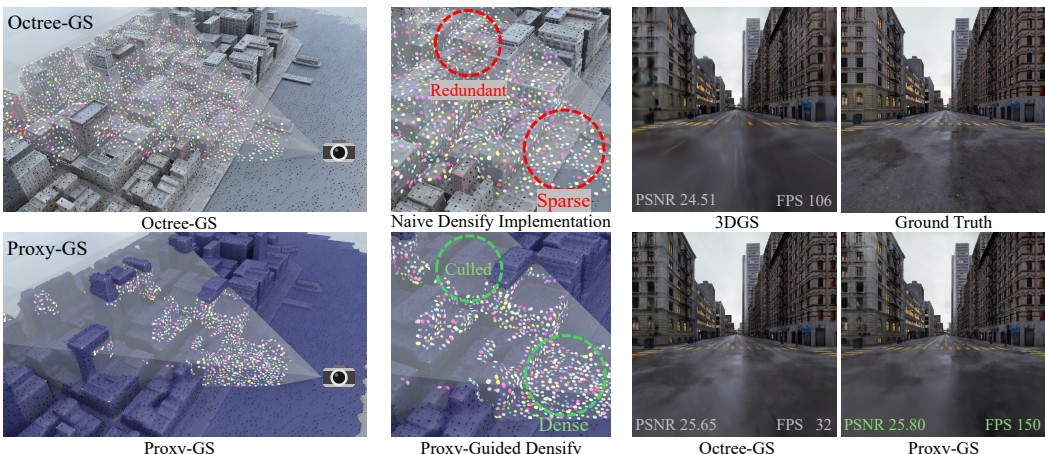

Figure 1: We propose **Proxy-GS**, an occlusion-aware training and inference framework built upon lightweight proxies. By introducing proxy-guided densification, our method effectively guides anchors to grow in more geometrically meaningful regions. As a result, Proxy-GS not only achieves higher rendering quality but also delivers significantly faster rendering compared to state-of-the-art MLP-based 3DGS approaches.

## ABSTRACT

3D Gaussian Splatting (3DGS) has emerged as an efficient approach for achieving photorealistic rendering. Recent MLP-based variants further improve visual fidelity but introduce substantial decoding overhead during rendering. To alleviate computation cost, several pruning strategies and level-of-detail (LOD) techniques have been introduced, aiming to effectively reduce the number of Gaussian primitives in large-scale scenes. However, our analysis reveals that significant redundancy still remains due to the lack of occlusion awareness. In this work, we propose Proxy-GS, a novel pipeline that exploits a proxy to introduce Gaussian occlusion awareness from any view. At the core of our approach is a fast proxy system capable of producing precise occlusion depth maps at resolution $1000\times1000$ under $1\,\mathrm{ms}$. This proxy serves two roles: first, it guides the culling of anchors and Gaussians to accelerate rendering speed. Second, it guides the densification towards surfaces during training, avoiding inconsistencies in occluded regions, and improving the rendering quality. In heavily occluded scenarios, such as the MatrixCity Streets dataset, Proxy-GS not only equips MLP-based Gaussian splatting with stronger rendering capability but also achieves faster rendering speed than the original 3DGS. Specifically, it achieves more than $2.5\times$ speedup over Octree-GS, and consistently delivers substantially higher rendering quality. Code will be public upon acceptance.

## 1 INTRODUCTION

With the emergence of Neural Radiance Fields (NeRF) (Mildenhall et al., 2020), high-quality novel view synthesis has become possible, but the slow rendering speed limits its practical use. Recently, 3D Gaussian Splatting (3DGS) (Kerbl et al., 2023b) has significantly improved efficiency, greatly advancing AR and VR applications. However, vanilla 3DGS often produces heavily redundant

Gaussians that attempt to fit every training view while neglecting the underlying scene geometry. To address this limitation and pursue higher-fidelity representations, structured MLP-based Gaussian approaches such as scaffold-GS (Lu et al., 2024) and Octree-GS (Ren et al., 2024) have been introduced.

At the core of the MLP-based 3DGS method lies an MLP decoder that conditions on the camera viewing direction to dynamically generate Gaussian attributes. Although these structured Gaussian variants substantially strengthen the modeling of challenging and view-dependent details, they also impose extra decoding operations at inference, leading to increased computational cost. This drawback becomes particularly critical in large-scale scene reconstruction, where the number of Gaussian primitives and rendering complexity grow dramatically, making efficient decoding and rendering indispensable.

Although pruning strategies (Fan et al., 2024; Lee et al., 2024; Liu et al., 2025) can be introduced to reduce redundancy, they inevitably lead to a loss in rendering quality. Meanwhile, following works (Ren et al., 2024; Kerbl et al., 2024; Cui et al., 2024) employ a level-of-detail (LOD) structure to mitigate redundancies from distant scene contents, but this approach is mainly effective in relatively occlusion-free environments. In contrast, real-world scenarios are full of occlusions, especially in large-scale modern city streets and complex indoor environments with multiple rooms. For future ultra-large VR walkthroughs that seamlessly span from indoor to outdoor scenes, effective occlusion culling becomes an essential and intuitive requirement.

Moreover, since most practitioners rely on consumer-grade GPUs rather than datacenter-oriented ones such as A100s, it is important to consider the hardware characteristics of these devices. Consumer GPUs, typically designed for gaming and graphics applications, are equipped with dedicated hardware rasterization units. The widespread adoption of 3DGS thus requires careful adaptation to hardware rasterization (fas, 2024).

To address the above limitations, we propose **Proxy-GS**, a proxy-guided Gaussian representation that leverages lightweight proxy meshes obtained through dedicated design. By bridging hardware rasterization with a PyTorch-based proxy renderer, Proxy-GS can efficiently cull occluded anchors with negligible time consumption and seamlessly integrate this process with the original frustum selection strategy. Furthermore, during training, the proxy guidance is incorporated again to provide stronger structural cues for anchor selection and densification.

As shown in Fig. 1, Proxy-GS not only achieves up to a $3\times$ speedup in rendering on top of existing MLP-based LOD frameworks Octree-GS (Ren et al., 2024) but also improves occlusion awareness in anchor selection, leading to higher rendering quality. Our main contributions can be summarized as follows:

- We design a **proxy-guided training pipeline** that incorporates structural priors from proxy meshes, enabling MLP-based approaches to be occlusion-aware and achieve higher rendering quality.

- Under a consistent training and testing setting, Proxy-GS achieves more than a $3\times$ FPS speedup over the LOD baseline on occlusion-rich scenes, while simultaneously improving rendering quality.

- We leverage engineering optimizations to reduce the time of acquiring a $1000^2$-resolution depth map to under $1\,\mathrm{ms}$.

## 2 RELATED WORK

### 2.1 NEURAL RENDERING

Neural Radiance Fields (NeRFs) (Mildenhall et al., 2020) pioneered the idea of representing a scene as a volumetric radiance field, enabling high-quality novel view synthesis for bounded scenes, typically centered around a single object. Subsequent extensions improved the scalability and visual fidelity of NeRF-based methods: Mip-NeRF (Barron et al., 2021) introduced proper anti-aliasing to handle multi-scale observations, NeRF++ (Zhang et al., 2020) lifted the constraint of strictly bounded scenes, and Mip-NeRF 360 (Barron et al., 2022) extended anti-aliased representations to

unbounded, object-centric settings. Despite these advances, NeRF-style volumetric rendering remains computationally expensive due to the need for dense ray sampling and neural field evaluation.

To overcome this inefficiency, 3D Gaussian Splatting (3DGS) (Kerbl et al., 2023a) was recently proposed as an explicit point-based alternative. By representing a scene with a set of anisotropic 3D Gaussian primitives and employing a splatting-based rasterization pipeline, 3DGS enables real-time rendering while preserving high visual quality. This paradigm shift bridges the gap between neural radiance fields and traditional graphics pipelines, offering both efficiency and scalability. While 3DGS achieves real-time performance with explicit Gaussian primitives, its reliance on directly optimized parameters often leads to limited expressiveness, particularly in capturing fine-grained appearance details and complex view-dependent effects. To address this shortcoming, MLP-based extensions such as Scaffold-GS (Lu et al., 2024) and Octree-GS (Ren et al., 2024) introduce neural decoders that generate Gaussian attributes from learned anchor features. By leveraging structured anchors and neural decoding, these approaches significantly improve the representational capacity, enabling more accurate modeling of geometry and appearance in large and challenging scenes.

However, this enhanced expressiveness comes at the cost of efficiency. The dependence on per-anchor MLP decoding introduces substantial computational overhead during inference, making rendering speed a critical bottleneck. Even with level-of-detail (LOD) designs to reduce the number of anchors processed per view, MLP-based methods still struggle to balance quality and efficiency, especially in large-scale urban or indoor environments with heavy occlusions. To this end, we are the first to address such occlusion-induced redundancy through a lightweight proxy mechanism.

## 2.2 Toward Faster 3D Gaussian Splatting Rendering

For rendering acceleration, many studies (Lee et al., 2024; Fan et al., 2024; Liu et al., 2025; Wang & Xu, 2025) have explored pruning or compression strategies to reduce the number of Gaussians and thus alleviate computational overhead. While such pruning-based methods can be effective to some extent, they inevitably face scalability bottlenecks in large scenes, where aggressive pruning results in performance degradation. Beyond pruning strategy, another line of research focuses on architectural designs for rendering acceleration. Among them, level-of-detail (LOD) architectures have become particularly influential. Hierarchical-GS (Kerbl et al., 2024) merges neighboring Gaussians to reduce rendering cost, achieving higher frame rates at the expense of some visual fidelity. LetsGo (Cui et al., 2024) jointly optimizes multi-resolution Gaussian models and demonstrates strong performance in LiDAR-based scenarios, yet its reliance on multi-resolution point cloud inputs incurs substantial training overhead and creates a strong dependence on point cloud accuracy. CityGaussian (Liu et al., 2024) further combines pruning strategies (Fan et al., 2024) with LOD-based rendering to enhance scalability in urban scenes.

While the aforementioned works improve efficiency for explicit 3DGS, LOD mechanisms have also been extended to MLP-based Gaussians. Octree-GS (Ren et al., 2024) organizes anchors into a multi-level octree, where the level selection is determined by the distance to the camera, thereby reducing the number of anchors decoded at each frame. This strategy alleviates part of the computational burden in large-scale scenes, but the rendering speed still leaves considerable room for improvement. Recent work Cache-GS (Tao et al., 2025) provides further acceleration by reusing decoded Gaussians, effectively doubling the rendering speed of Octree-GS, although this comes with a noticeable loss in rendering quality. In parallel, methods like FLASH-GS (Feng et al., 2024) target low-level CUDA optimizations of the original 3DGS pipeline, aiming to improve efficiency at the kernel level. Recent work has also explored leveraging occlusion for accelerating rendering. For example, Ye et al. (2025) proposed using pre-rendered depth maps to guide 3DGS rendering. However, their depth acquisition relies on 2DGS rendering, which is less efficient compared to our lightweight proxy-based approach.

## 3 Preliminaries

### 3.1 MLP-based 3DGS

To exploit the structural priors provided by Structure-from-Motion (SfM), a line of work such as Scaffold-GS (Lu et al., 2024) and Octree-GS (Ren et al., 2024) has been developed. Instead of reconstructing Gaussians directly from sparse SfM points, Scaffold-GS first builds a coarse voxel

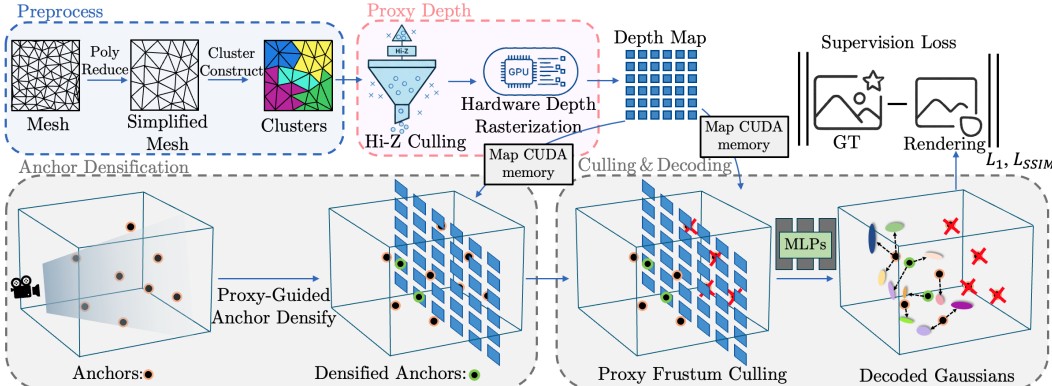

Figure 2: **Proxy-GS Framework.** We first construct a lightweight proxy mesh. During rendering, hardware rasterization produces a depth map in under 1 ms, which is then used to efficiently cull anchors that are occluded. During training, in addition to the same rendering pipeline, we further introduce structure-aware anchor densification, encouraging anchors to grow adaptively along the proxy mesh geometry.

grid and places anchor points at the voxel centers. Each anchor is associated with a latent feature vector $f$, which is fed into a multi-layer perceptron (MLP) to decode the corresponding Gaussian attributes:

$$\{\mu_j, \Sigma_j, c_j, \alpha_j\}_{j \in \mathcal{M}} = \text{MLP}_\theta(f_i, v_i)_{i \in \mathcal{N}}, \tag{1}$$

where $\theta$ denotes the MLP parameters, and $\mu_j$, $\Sigma_j$, $c_j$, and $\alpha_j$ represent the mean, covariance, color, and opacity of the $j$-th Gaussian derived from the $i$-th anchor under viewing direction $v_i$. The generated neural Gaussians are subsequently rasterized in the same way as explicit 3D Gaussians.

The advantage of anchor-based placement is that the decoded Gaussians inherit structural cues from the underlying SfM prior, which reduces redundancy and improves robustness for novel view rendering. Octree-GS extends this framework by substituting the voxel grid with an explicit octree representation, enabling the scene to be modeled at multiple resolutions.

The hierarchical design of the octree naturally supports level-of-detail (LOD) construction. During rendering, appropriate LOD levels can be selected adaptively based on the camera distance, thereby reducing decoding cost and improving scalability to larger-scale scenes.

## 3.2 HARDWARE RASTERIZATION

Hardware rasterization denotes the GPU's fixed/near–fixed-function graphics path that transforms vertices to clip/NDC, discretizes primitives into fragments, interpolates attributes, and resolves visibility via depth/stencil tests and blending before writing to render targets. This behavior is standardized in modern graphics APIs and is executed by specialized units. The pixel backend, commonly called the Raster Operations Processor (ROP, a.k.a. render output unit) houses depth/stencil units that perform depth and stencil tests and update the corresponding buffers, and color units that handle blending, format conversion/MSAA resolves, and render-target writes. These mechanisms underpin the extreme throughput and bandwidth efficiency of the pipeline. Architecturally, rasterization evolved from fixed-function to programmable/unified shader models and is realized across immediate-mode and tile/binning GPU designs, but the visibility tests and depth buffering remain conceptually consistent. In this work, we will later exploit this machinery in a depth-only pass on a proxy mesh to obtain a conservative Z-buffer at negligible cost, which we then consume as a visibility prior.

## 4 METHOD

### 4.1 MOTIVATION

Reconstructing large-scale scenes with high occlusion presents unique challenges due to the vast number of Gaussians and anchors involved. As illustrated in Fig. 1, When visualizing the anchors used for decoding, we observe a significant mismatch between the decoded anchors and those that are intuitively required for accurate rendering. In particular, a large proportion of anchors correspond to heavily occluded regions, which substantially increases the decoding burden without contributing to the final image quality. Effective occlusion culling, therefore, has the potential to greatly reduce computational cost.

Existing MLP-based works, such as Octree-GS Ren et al. (2024) and Scaffold-GS Lu et al. (2024), design anchor structures to better exploit the inherent hierarchy and structural priors. However, since their anchor selection does not explicitly account for occlusions, the anchors are optimized merely to fit RGB images. As a result, the binding between anchors and their associated Gaussians can become inconsistent in space, leading to redundant decoding and degraded structural interpretability

### 4.2 PROXY GUIDED FILTER

A central question in our study is how to obtain occlusion relationships both efficiently and with negligible loss of accuracy. We find that leveraging lightweight proxy meshes for hardware rasterization enables depth rendering at only a marginal time cost. For many outdoor large-scale scenes, dense point clouds are already available or can be generated using tools such as COLMAP. In contrast, indoor scenes often contain texture-less regions that cause SfM-based reconstruction to fail, so we adopt a hybrid strategy. Specifically, we combine state-of-the-art monocular depth estimation Wang et al. (2025) with PGSR Chen et al. (2024), in a manner similar to recent indoor surface reconstruction approaches Zhang et al. (2024); Ren et al. (2025). Further implementation details are provided in the Appendix A.4. We construct proxy meshes using existing engineering pipelines and apply surface simplification to retain only coarse geometric structures. This proxy is sufficient to fully exploit the high throughput of hardware fixed-function units for efficient depth generation.

To further accelerate the process, the scene is partitioned into fine-grained clusters, and hierarchical visibility checks such as Hierarchical Z-buffer (Hi-Z) culling Greene et al. (1993) are employed to quickly cull invisible clusters. In the fragment stage, early-fragment tests (Early-Z) are enabled, and we keep the fragment shader minimal by removing operations unrelated to depth writes. This allows our method to output depth maps at a high speed even in complex and large-scale urban scenes, as shown in Fig. 3. The depth map is kept on GPU and directly exploited in CUDA occlusion culling to avoid GPU-CPU-GPU round-trip overhead. For more details, please refer to the Appendix A.5.

Then we fuse the occlusion culling and frustum culling of anchors in a single CUDA kernel: Given an original anchor point $\mathbf{p}_{\text{orig}} = (x, y, z) \in \mathbb{R}^3$, the point is first transformed into the camera (view) coordinate system via the view matrix $V \in \mathbb{R}^{4\times4}$:

$$\mathbf{p}_{\text{view}} = V \begin{bmatrix} \mathbf{p}_{\text{orig}} \\ 1 \end{bmatrix}. \tag{2}$$

Then it transfers to the homogeneous clip space using the projection matrix $P \in \mathbb{R}^{4\times4}$:

$$\mathbf{p}_{\text{hom}} = P\mathbf{p}_{\text{view}} = (x_h, y_h, z_h, w_h)^\top. \tag{3}$$

To obtain normalized device coordinates (NDC), we divide by the homogeneous component:

$$\mathbf{p}_{\text{ndc}} = \left( \frac{x_h}{w_h + \epsilon}, \frac{y_h}{w_h + \epsilon}, \frac{z_h}{w_h + \epsilon} \right), \quad \epsilon = 10^{-7}. \tag{4}$$

We denote the resulting coordinates as $(x_{\text{ndc}}, y_{\text{ndc}}, z_{\text{ndc}})$.

A visibility check is then performed: points with $z_h \leq \tau, \quad \tau = 10^{-4}$, are regarded as invalid (filtered), since they lie behind the camera or are too close to the near plane. After projecting to normalized device coordinates (NDC), we map the coordinates to discrete pixel indices $(u, v)$:

$$x_{\text{pix}} = \left\lfloor \frac{(x_{\text{ndc}} + 1)}{2} \cdot W \right\rfloor, \quad y_{\text{pix}} = \left\lfloor \frac{(y_{\text{ndc}} + 1)}{2} \cdot H \right\rfloor, \tag{5}$$

where $W$ and $H$ denote the image width and height, respectively. A pixel is discarded if it falls outside the image boundary:

$$x_{\text{pix}} < 0 \ \vee \ x_{\text{pix}} \geq W \ \vee \ y_{\text{pix}} < 0 \ \vee \ y_{\text{pix}} \geq H. \tag{6}$$

For valid pixels, we retrieve the hardware depth $z_{hw} \in [0, 1]$ at $(x_{\text{pix}}, y_{\text{pix}})$ from the depth image. We then convert it to the *linear* camera-space depth using the near/far planes $n, f$:

$$d_{\text{mesh}}(x_{\text{pix}}, y_{\text{pix}}) = \frac{n\,f}{f - z_{hw}(x_{\text{pix}}, y_{\text{pix}})\,(f - n)}. \tag{7}$$

Finally, we apply a small safety margin $\gamma$:

$$\hat{d}(x_{\text{pix}}, y_{\text{pix}}) = d_{\text{mesh}}(x_{\text{pix}}, y_{\text{pix}}) + \gamma. \tag{8}$$

If the depth value is invalid, the point is not culled. Otherwise, we apply the depth test:

$$\text{Cull}(\mathbf{p}) = \begin{cases} \text{true}, & z_{\text{h}} > \hat{d}(x_{\text{pix}}, y_{\text{pix}}), \\ \text{false}, & z_{\text{h}} \leq \hat{d}(x_{\text{pix}}, y_{\text{pix}}). \end{cases} \tag{9}$$

To summarize, a point is removed if its camera-space depth lies *behind* the depth map at the corresponding pixel, which effectively performs occlusion culling on the image plane.

### 4.3 PROXY-GUIDED DENSIFICATION

In the original anchor-growing densification strategy, new anchors are generated around Gaussian splats that exhibit large gradients during training. However, this procedure may introduce redundant anchors behind the proxy mesh depth: although these Gaussians have large gradients, the newly grown anchors do not contribute to rendering due to occlusion.

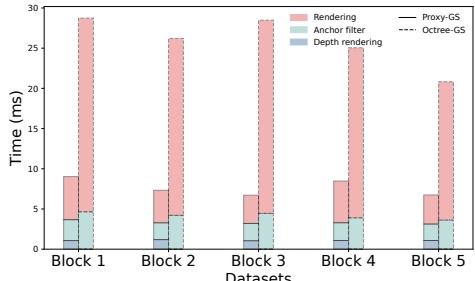

To tackle this limitation, and inspired by the multi-view depth densification strategy in Li et al. (2024), we introduce **proxy-guided densification**, which explicitly projects anchors onto the surface of the proxy mesh. Since proxy depth maps are pre-computed, we can measure the patch-wise L1 loss and identify regions where the rendering error is consistently large.

Figure 3: **Comparison of the time proportion of each inference component (Rendering, anchor filter, depth rendering) with that of Octree-GS.**

To achieve this, patches with abnormally high error are identified by comparing to the mean error $\bar{\ell}$ within the same frame. We compute the per-patch loss as the average of pixel losses:

$$\ell_{\mathcal{P}} = \frac{1}{|\Omega_{\mathcal{P}}|} \sum_{(u,v) \in \Omega_{\mathcal{P}}} \ell(u, v), \qquad \bar{\ell} = \frac{1}{|\mathcal{S}|} \sum_{\mathcal{P} \in \mathcal{S}} \ell_{\mathcal{P}}.$$

We select patches that satisfy

$$\ell_{\mathcal{P}} > \tau, \qquad \tau = 3\,\bar{\ell}.$$

For each selected patch $\mathcal{P}$, choose a representative pixel $(u_{\mathcal{P}}, v_{\mathcal{P}})$ (e.g., the patch center), read the hardware depth $z_h(u_{\mathcal{P}}, v_{\mathcal{P}})$, and convert it to linear camera-space depth with near/far $(n, f)$, to obtain $d_{\text{mesh}}(u_{\mathcal{P}}, v_{\mathcal{P}})$. We then back-project this pixel to 3D and take it as the new anchor position:

$$\hat{\mathbf{p}}_{\mathcal{P}} = \mathbf{o} + \mathbf{R}^{\top}\left(d_{\text{mesh}}(u_{\mathcal{P}}, v_{\mathcal{P}})\,\mathbf{K}^{-1}\begin{bmatrix} u_{\mathcal{P}} \\ v_{\mathcal{P}} \\ 1 \end{bmatrix}\right), \qquad \mathbf{a} \leftarrow \hat{\mathbf{p}}_{\mathcal{P}}.$$

To prevent redundancy in 3D space, we maintain a proxy-grid with cell size $h$ and origin $\mathbf{b}_{\min}$, and allow up to $K$ anchors per cell:

$$\mathbf{c}(\mathbf{a}) = \left\lfloor \frac{\mathbf{a} - \mathbf{b}_{\min}}{h} \right\rfloor \in \mathbb{Z}^3, \qquad \text{insert } \mathbf{a} \text{ if } \kappa[\mathbf{c}(\mathbf{a})] < K,$$

where $\kappa[\cdot] \in \mathbb{N}$ tracks the current number of anchors in each cell.

Table 1: Quantitative results on MatrixCity (Li et al., 2023). We report average results over Block 1&2, Block 3&4, and Block 5. (Block 1&2 and 3&4 represent the average evaluation metrics of their respective two blocks.) The **best** and second-best are highlighted.

| Methods | Block 1&2 | | | | Block 3&4 | | | | Block 5 | | | |
|---|---|---|---|---|---|---|---|---|---|---|---|---|
| | PSNR↑ | SSIM↑ | LPIPS↓ | FPS↑ | PSNR↑ | SSIM↑ | LPIPS↓ | FPS↑ | PSNR↑ | SSIM↑ | LPIPS↓ | FPS↑ |
| 3DGS (Kerbl et al., 2023a) | 21.55 | 0.730 | 0.366 | 115 | 20.78 | 0.739 | 0.372 | 114 | 20.70 | 0.697 | 0.425 | 121 |
| Scaffold-GS (Lu et al., 2024) | 21.44 | 0.721 | 0.375 | 81 | 20.56 | 0.727 | 0.376 | 66 | 20.56 | 0.693 | 0.426 | 71 |
| Hierarchical-GS (Kerbl et al., 2024) | 20.50 | 0.707 | 0.418 | 61 | 20.38 | 0.719 | 0.422 | 41 | 20.22 | 0.673 | 0.463 | 60 |
| Hierarchical-GS($\tau_1$) | 20.50 | 0.706 | 0.419 | 62 | 20.38 | 0.718 | 0.424 | 45 | 20.22 | 0.672 | 0.466 | 66 |
| Hierarchical-GS($\tau_2$) | 20.46 | 0.702 | 0.423 | 71 | 20.30 | 0.711 | 0.431 | 49 | 20.20 | 0.671 | 0.467 | 75 |
| Hierarchical-GS($\tau_3$) | 20.01 | 0.678 | 0.450 | 85 | 19.71 | 0.680 | 0.464 | 63 | 20.01 | 0.657 | 0.483 | 90 |
| Octree-GS (Ren et al., 2024) | 21.94 | 0.737 | 0.347 | 32 | 20.95 | 0.743 | 0.354 | 30 | 21.41 | 0.731 | 0.375 | 48 |
| Proxy-GS | **22.11** | **0.751** | **0.330** | **126** | **21.06** | **0.751** | **0.348** | **134** | **21.68** | **0.744** | **0.362** | **151** |

Table 2: Quantitative results on real world Outdoor and Indoor datasets (Xiong et al., 2024; Kerbl et al., 2024; Barron et al., 2023). **best** and second-best are highlighted.

| Methods | CUHK-LOWER | | | | Berlin | | | | Small City | | | |
|---|---|---|---|---|---|---|---|---|---|---|---|---|
| | PSNR↑ | SSIM↑ | LPIPS↓ | FPS↑ | PSNR↑ | SSIM↑ | LPIPS↓ | FPS↑ | PSNR↑ | SSIM↑ | LPIPS↓ | FPS↑ |
| 3DGS (Kerbl et al., 2023a) | 25.48 | 0.729 | 0.389 | 138 | 27.79 | 0.907 | 0.223 | 187 | 22.90 | 0.727 | 0.372 | 132 |
| Scaffold-GS (Lu et al., 2024) | 26.30 | 0.785 | 0.282 | 117 | 27.80 | 0.912 | **0.213** | 128 | 20.00 | 0.713 | 0.370 | 62 |
| Hierarchical-GS (Kerbl et al., 2024) | 25.18 | 0.707 | 0.408 | 90 | 27.65 | 0.902 | 0.228 | 145 | 22.07 | 0.728 | 0.377 | 89 |
| Hierarchical-GS($\tau_1$) | 25.19 | 0.708 | 0.408 | 82 | 27.65 | 0.901 | 0.229 | 150 | 22.07 | 0.728 | 0.377 | 90 |
| Hierarchical-GS($\tau_2$) | 25.14 | 0.705 | 0.411 | 96 | 27.60 | 0.899 | 0.232 | 152 | 22.07 | 0.728 | 0.378 | 106 |
| Hierarchical-GS($\tau_3$) | 24.58 | 0.678 | 0.435 | 120 | 27.34 | 0.890 | 0.244 | 160 | 22.02 | 0.722 | 0.386 | 119 |
| Octree-GS (Ren et al., 2024) | 26.42 | 0.794 | 0.267 | 212 | 27.83 | 0.911 | 0.218 | 263 | 23.03 | 0.731 | 0.355 | 51 |
| Proxy-GS | **26.44** | **0.795** | **0.262** | **239** | **27.85** | **0.912** | 0.216 | **275** | **23.09** | **0.736** | **0.344** | **139** |

## 5 EXPERIMENT

**Datasets.** We begin by comparing our approach with other methods on the large-scale urban dataset (Li et al., 2023) to assess rendering quality. We follow the partition script of the MatrixCity, and divided the 8477 street images in its Small City into 5 blocks. Details can be seen in the Appendix A.1. The evaluation is further extended to large-scale indoor scenes from Zip-NeRF (Barron et al., 2023). In addition, we also test on real-world street scenes from the Small City dataset (Kerbl et al., 2024), as well as real-world aerial-view scenes from CUHK-LOWER (Xiong et al., 2024), which contain relatively fewer occlusions.

**Evaluation Criterion.** We adopt three widely used image quality metrics to evaluate novel view synthesis: peak signal-to-noise ratio (PSNR), structural similarity index (SSIM), and learned perceptual image patch similarity (LPIPS) (Zhang et al., 2018). In addition, we report frames per second (FPS) to measure the rendering efficiency of different methods.

**Implementation Details.** Our method is implemented on top of the state-of-the-art MLP-based Octree-GS (Ren et al., 2024), following its default initialization and LOD strategy. For comparison, we also re-implement 3DGS (Kerbl et al., 2023a), Scaffold-GS (Lu et al., 2024), and Hierarchical-GS (Kerbl et al., 2024), and train all methods for 40k iterations. Specifically, for the evaluation of Hierarchical-GS, we set the $\tau_1, \tau_2, \tau_3 = 3, 6, 15$. For approaches that do not employ MLPs, such as 3DGS and Hierarchical-GS, their default configurations typically yield higher rendering FPS but exhibit a noticeable quality gap compared to Octree-GS. Since an increased number of Gaussian primitives generally leads to better rendering quality (Zhao et al., 2024), we reduce the densification threshold to $10^{-4}$ across all scenes to ensure a fair comparison, resulting in rendering quality closer to that of Octree-GS. Unlike Octree-GS, Scaffold-GS initializes with fewer anchors due to the absence of multi-round sampling. To improve its rendering fidelity, we adopt a smaller voxel size of $10^{-4}$ together with a lower densification threshold of $10^{-4}$. All training experiments are performed on a single NVIDIA A100-40GB GPU. For inference, we employ a consumer-grade RTX 4090 GPU to reflect real-world deployment scenarios better.

### 5.1 MAIN RESULTS

**Novel View Synthesis and rendering FPS.** As shown in Tab. 1 and Tab. 2, our method achieves higher or comparable rendering quality compared to all other baselines. Moreover, Fig. 4 illustrates

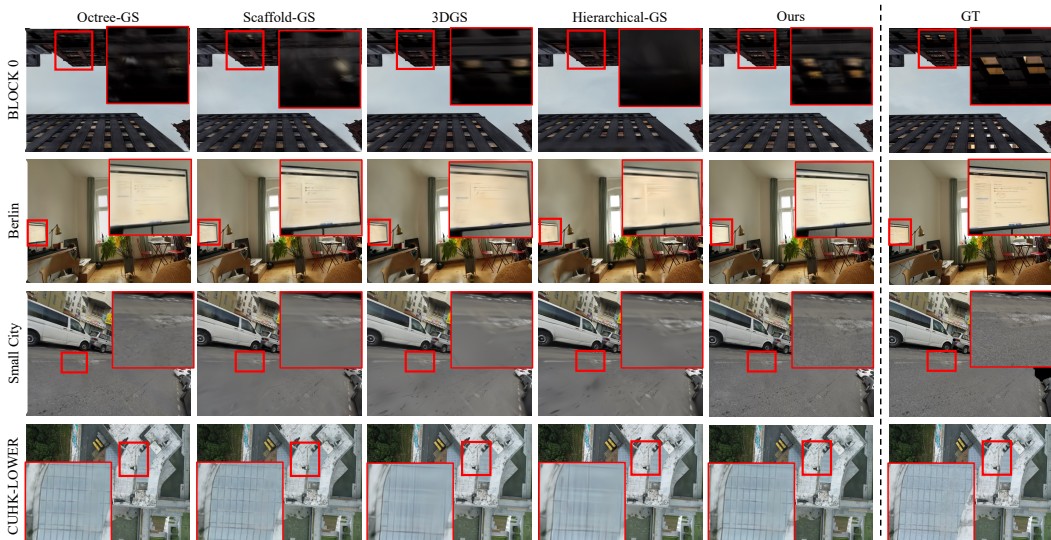

Figure 4: **Qualitative comparison.** Visualization on different datasets (Li et al., 2023; Xiong et al., 2024; Barron et al., 2023; Kerbl et al., 2024).

that our approach better preserves fine details such as building windows and crosswalk patterns. In particular, as shown in Tab. 1, the large urban street scenes simulated in MatrixCity are highly suited to our approach, where we consistently outperform existing methods in both rendering quality and speed.

Furthermore, to demonstrate the generality of our method, in Tab. 2 We also evaluate our method on aerial-view scenes, indoor environments, and real-world town streets, where it achieves comparable or superior performance against current state-of-the-art methods. Although aerial scenes typically involve limited occlusions and the current indoor dataset often contains relatively few rooms with sparse occlusion patterns, our method still yields noticeable improvements. Moreover, for small-city street scenes, which bear resemblance to the MatrixCity dataset, our approach delivers substantial improvements over the MLP-based method Octree-GS, achieving higher rendering quality while boosting FPS by nearly $3\times$. These results collectively demonstrate the broad applicability of our method across diverse scenarios, while also highlighting that the extent of performance gains may vary depending on the characteristics of the scene.

## 5.2 ABLATIONS

**Effect of training procedure.** As shown in Tab. 4, we conduct ablation studies on different training strategies. **ID 1** corresponds to the default Octree-GS training and testing pipeline, which serves as our baseline. **ID 2** applies our proxy-guided rendering strategy at test time only, without modifying the Octree-GS training process. Although this setting brings more than a $3\times$ FPS increase, the inconsistency between anchors and their associated Gaussians during training leads to a noticeable

Table 3: **Ablations of different safety margin of depth culling $\gamma$ trained on Small City Kerbl et al. (2024).**

| $\gamma$ | PSNR↑ | SSIM↑ | LPIPS↓ | FPS ↑ |
|---|---|---|---|---|
| 0.1 | 22.94 | 0.734 | 0.349 | 142 |
| 0.3 | 23.09 | 0.736 | 0.344 | 139 |
| 0.6 | 23.02 | 0.735 | 0.348 | 135 |
| 1.0 | 23.05 | 0.736 | 0.345 | 128 |

drop in rendering quality. **ID 3** further enforces consistency by employing proxy-guided rendering also during training. In this case, rendering quality surpasses the baseline, while FPS slightly decreases compared to **ID 2**, mainly because more anchors grow before being culled by occlusion.

**ID 4** incorporates the proposed proxy-guided densification strategy in addition to proxy-guided training and rendering. This setting achieves the best balance, delivering further improvements in rendering quality while maintaining a comparable FPS to **ID 3**.

**Rendering time analysis.** In Fig. 3, we quantify the proportion of inference time spent on each component. The lightweight proxy-based depth rendering takes nearly negligible time (around

Table 4: **Ablations of different training and inference strategies on Block 5.** Average anchor denotes the average number of decoded anchors in the scene.

| ID | Occlusion Training | Proxy-guided Densification | Proxy-guided Inference | PSNR↑ | FPS↑ | Average anchor |
|---|---|---|---|---|---|---|
| 1 | ✗ | ✗ | ✗ | 21.41 | 48 | 719k |
| 2 | ✗ | ✗ | ✓ | 19.06 | 165 | 82k |
| 3 | ✓ | ✗ | ✓ | 21.50 | 147 | 93k |
| 4 | ✓ | ✓ | ✓ | 21.68 | 143 | 106k |

1 ms). Our anchor filtering is also faster due to the reduced number of anchors. The rendering stage is where most of the savings come from: with fewer anchors, both the decoding overhead and Gaussian rasterization are significantly reduced. For more details, we also record the average decode anchors in the Appendix A.3.

**Integration with different 3DGS renderers.** Since our method primarily optimizes anchors and thus indirectly reduces the number of rendered Gaussians, it can be naturally combined with existing acceleration techniques for the original 3DGS to achieve even higher speed. In Table 5, we evaluate on Block 1. Here, Original 3DGS denotes the default renderer used in Proxy-GS. Replacing it with FlashGS brings a minor improvement, while using a hardware rasterizer for 3DGS slightly compromises rendering quality but further boosts the frame rate by nearly 40 FPS.

Table 5: **Integration with different 3DGS rendering accelerations.** We evaluate our method combined with existing approaches (Feng et al., 2024; fas, 2024) on Block 1.

| Method | PSNR↑ | SSIM↑ | LPIPS↓ | FPS↑ |
|---|---|---|---|---|
| Original 3DGS | 23.27 | 0.786 | 0.322 | 112 |
| FlashGS | 23.27 | 0.785 | 0.322 | 115 |
| Hardware 3DGS | 23.20 | 0.781 | 0.328 | 155 |

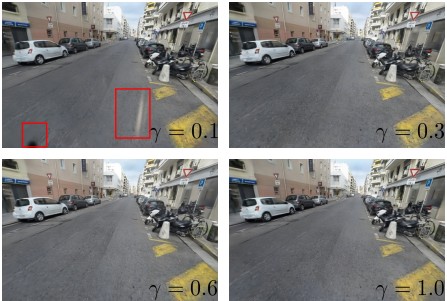

Figure 5: **Visualization on different safety margins.**

**Safety margin of the occlusion culling.** In the Appendix A.2, we report all the results with the hardware 3DGS as the default renderer.

In Tab. 3, we report results on the Small City dataset by varying the depth culling threshold $\gamma$ in Eq. 8. We observe that $\gamma = 0.3$ yields the best trade-off between rendering quality and speed. As can be seen in Fig. 5, when the threshold is too small $\gamma = 0.1$, it leads to rendering artifacts in nearby regions. However, setting $\gamma$ too large is also undesirable: a larger threshold introduces excessive anchors, which increases structural redundancy and reduces FPS, while a too small threshold restricts anchor growth and degrades rendering quality.

## 6 CONCLUSION

In this work, we propose Proxy-GS, a proxy-guided training and inference framework for MLP-based 3D Gaussian Splatting. Our carefully designed proxy-guided filter enables nearly lossless depth acquisition and occlusion culling, while the proxy-guided densification effectively leverages geometric priors from proxies to provide a more structured densification mechanism. Extensive experiments demonstrate that our framework consistently improves both rendering quality and efficiency across diverse scenarios. In particular, on occlusion-rich scenes, Proxy-GS achieves up to a 2.5× speedup, significantly advancing the practicality of MLP-based methods for VR/AR applications, and establishing a new state-of-the-art in efficient 3D scene representation.

REPRODUCIBILITY

To ensure reproducibility, we will release the complete training and inference code. All results reported in this paper can be reproduced using the released repository, along with the same experimental settings described in the main text and Appendix. The final model of all the datasets will also be made publicly available.

ETHICS STATEMENT

Our research is devoted to enhancing the efficiency and rendering quality of 3D reconstruction techniques. No experiments involve human participants, personally identifiable information, or sensitive content. The datasets employed are openly released for academic purposes and have been broadly utilized in prior literature. We adhered to their licensing terms and conducted all experiments in a manner consistent with data privacy and integrity. We consider this work to present no evident ethical concerns or potential societal risks.

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

# A APPENDIX

## A.1 DIVISION DETAIL ON MATRIXCITY

We divide all the Horizon street scenes in MatrixCity's small city into five blocks (eg. Block 1, Block 2), The partition margin details is in Tab. 6

## A.2 COMBINE WITH HARDWARE 3DGS

We combine our method with Hardware 3DGS (fas, 2024) in Tab. 7 and Tab. 8. As observed, the FPS improves across all datasets, but due to the precision settings used, there is a noticeable decline in rendering quality.

Table 7: Combine with Hardware 3DGS (fas, 2024), quantitative results on MatrixCity (Li et al., 2023)

| Methods | Block 1&2 | | | | Block 3&4 | | | | Block 5 | | | |
|---|---|---|---|---|---|---|---|---|---|---|---|---|
| | PSNR↑ | SSIM↑ | LPIPS↓ | FPS↑ | PSNR↑ | SSIM↑ | LPIPS↓ | FPS↑ | PSNR↑ | SSIM↑ | LPIPS↓ | FPS↑ |
| Proxy-GS | **22.11** | **0.751** | **0.330** | 126 | **21.06** | **0.751** | **0.348** | 134 | **21.68** | **0.744** | **0.362** | 151 |
| +Hardware 3DGS (fas, 2024) | 22.05 | 0.747 | 0.338 | **167** | 20.86 | 0.743 | 0.357 | **174** | 21.58 | 0.735 | 0.372 | **196** |

Table 8: Combine with Hardware 3DGS (fas, 2024), quantitative results on real world Outdoor and Indoor datasets (Xiong et al., 2024; Kerbl et al., 2024; Barron et al., 2023).

| Methods | CUHK-LOWER | | | | Berlin | | | | Small City | | | |
|---|---|---|---|---|---|---|---|---|---|---|---|---|
| | PSNR↑ | SSIM↑ | LPIPS↓ | FPS↑ | PSNR↑ | SSIM↑ | LPIPS↓ | FPS↑ | PSNR↑ | SSIM↑ | LPIPS↓ | FPS↑ |
| Proxy-GS | 26.44 | 0.795 | 0.262 | 239 | 27.85 | 0.912 | 0.216 | 275 | 23.09 | 0.736 | 0.344 | 139 |
| +Hardware 3DGS (fas, 2024) | 26.28 | 0.787 | 0.265 | 280 | 27.78 | 0.906 | 0.210 | 325 | 22.91 | 0.732 | 0.343 | 163 |

## A.3 AVERAGE DECODED ANCHOR NUMBER ON ALL THE DATASETS

. In Tab. 9, we report the average number of anchors used during training and inference across all datasets. It can be observed that our method consistently reduces the decoding burden, although the degree of improvement varies across different scenes.

Table 6: Partition information in MatirxiCity.

| Block | $x_{\min}$ | $x_{\max}$ | $y_{\min}$ | $y_{\max}$ |
|---|---|---|---|---|
| 1 | −9.80 | −2.64 | 0 | 3.9 |
| 2 | −2.64 | 0.44 | 0 | 3.9 |
| 3 | 0.44 | 3.52 | 0 | 3.9 |
| 4 | 3.52 | 8.70 | 0 | 3.9 |
| 5 | −6.90 | 6.90 | 3.9 | 7.4 |

## A.4 MESH
EXTRACTION ON DIFFERENT DATASETS

### A.4.1 INDOOR AND OUTDOOR SCENES WITH DENSE POINT CLOUDS

We describe the mesh extraction process when dense point clouds are available for both indoor and outdoor environments. This category includes real-world datasets that provide LiDAR point clouds (e.g., Xiong et al. (2024)), where mesh generation can be directly performed using surface reconstruction methods, such as (Huang et al., 2023). In addition, for synthetic datasets such as MatrixCity, ground-truth depth maps are available, which can be fused via TSDF to obtain high-quality meshes.

### A.4.2 INDOOR SCENES WITH SPARSE COLMAP POINT CLOUDS

We describe the workflow of mesh extraction in indoor scenes where only sparse COLMAP reconstructions are available. Directly relying on COLMAP to generate dense point clouds in indoor environments is often unreliable, as such scenes frequently contain large textureless regions. To address this challenge, we follow the recent advances in 3DGS-based indoor reconstruction methods (Ren et al., 2025; Zhang et al., 2024), where sparse texture cues are complemented by state-of-the-art monocular depth estimation (e.g., MoGe2 (Wang et al., 2025)) and further refined with multi-view constraints from PGSR Chen et al. (2024). This hybrid strategy enables the recovery of reasonable indoor meshes despite the limitations of sparse COLMAP input.

Table 9: Average anchor number used to decode all the datasets

| Method | Block 1&2 | Block 3&4 | Block 5 | Berlin | CUHK-LOWER | Small City |
|---|---|---|---|---|---|---|
| Proxy-GS | 190k | 190k | 80k | 40k | 110k | 350k |
| Octree-GS | 800k | 1040k | 720k | 60k | 120k | 840k |

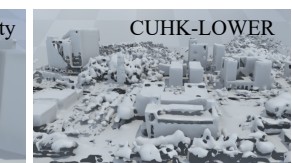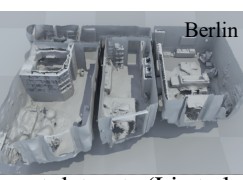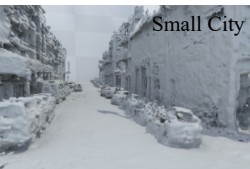

Figure 6: **Mesh visualization**. Scenes include different datasets (Li et al., 2023; Xiong et al., 2024; Barron et al., 2023; Kerbl et al., 2024).

### A.4.3 OUTDOOR SCENES WITH SPARSE COLMAP POINT CLOUDS

For outdoor environments where the reconstruction relies solely on sparse COLMAP point clouds, the abundance of feature points generally mitigates the issue of sparse textures. However, due to the large spatial extent, many 3DGS-based indoor reconstruction methods encounter out-of-memory (OOM) problems when applied to outdoor scenes. To address this, we employ CityGS-X (Gao et al., 2025), a state-of-the-art large-scale geometric reconstruction framework, which leverages multi-GPU parallelism to achieve scalable mesh generation with competitive performance.

### A.4.4 MESH VISUALIZATION

As shown in Fig. 6, we visualize all the lightweight proxies. Our method does not require highly accurate meshes; an approximate geometry is sufficient. Thanks to the anchor-based filtering, the subsequent growth of Gaussians introduces offsets that provide additional tolerance, thereby ensuring that our approach maintains a certain degree of robustness to mesh inaccuracies.

## A.5 FAST DEPTH ACQUISITION

### A.5.1 OVERVIEW.

We follow a modern real-time rendering pipeline to obtain high-quality depth maps at minimal latency. The key ideas are: (i) *preprocess* the reconstructed mesh into compact *clusters*; (ii) perform fully GPU-resident *frustum* and *hierarchical-Z (Hi-Z)* occlusion culling at *cluster* granularity each frame; (iii) emit a *depth-only* pass that leverages Early-Z; and (iv) *zero-copy* the resulting depth buffer into the learning runtime (PyTorch) via Vulkan–CUDA interop, avoiding CPU round trips. This section details each component.

### A.5.2 PREPROCESSING: FROM RECONSTRUCTED MESH TO CLUSTERS.

Given a triangle mesh $\mathcal{M} = (\mathcal{V}, \mathcal{F})$ obtained by the reconstruction routine above, we apply the following:

1. **Topology-preserving simplification.** We reduce face count with a quadric-error-metric (QEM) style simplifier while enforcing feature and boundary preservation. For a vertex in homogeneous coordinates $\tilde{\mathbf{x}} = (x, y, z, 1)^\top$ and its incident face planes $\{\mathbf{p}_f = (a, b, c, d)^\top\}$ (with $\|(a, b, c)\|_2 = 1$ for all $f$), the local quadric is

$$Q = \sum_f \mathbf{p}_f \mathbf{p}_f^\top,$$

These per-vertex quadrics are *accumulated* and then used by an edge-collapse procedure to decide the contraction position and cost, which removes superfluous micro-triangles commonly produced by reconstruction and improves cache locality and GPU occupancy.

**Edge-collapse simplification with QEM.** For each vertex $v$, accumulate $Q_v = \sum_{f \in N(v)} \mathbf{p}_f \mathbf{p}_f^\top$. To collapse an edge $(i, j)$, combine quadrics

$$Q' = Q_i + Q_j, \qquad E(\tilde{\mathbf{x}}) = \tilde{\mathbf{x}}^\top Q' \tilde{\mathbf{x}}.$$

Partition $Q'$ as $Q' = \begin{bmatrix} A & \mathbf{b} \\ \mathbf{b}^\top & c \end{bmatrix}$ with $A \in \mathbb{R}^{3 \times 3}$, $\mathbf{b} \in \mathbb{R}^3$, $c \in \mathbb{R}$. The optimal contraction position is

$$\mathbf{x}^* = \underset{\mathbf{x} \in \mathbb{R}^3}{\arg\min} \ \mathbf{x}^\top A \mathbf{x} + 2\,\mathbf{b}^\top \mathbf{x} + c = -A^{-1}\mathbf{b} \quad \text{(if } A \text{ is invertible)},$$

with cost $\delta = E([\mathbf{x}^{*\top}, 1]^\top)$.

If $A$ is singular, evaluate $\{\mathbf{x}_i, \mathbf{x}_j, (\mathbf{x}_i + \mathbf{x}_j)/2\}$ and pick the one with minimal $E$. We maintain a priority queue keyed by $\delta$ and iteratively collapse the lowest-cost edge, updating connectivity and setting the new vertex quadric to $Q'$. Collapses that would break manifoldness or flip triangle orientations are forbidden.

**Boundary/feature preservation.** For a boundary or sharp-crease edge with unit tangent $\mathbf{t}$ and unit average normal $\hat{\mathbf{n}}$, add two *constraint planes* whose intersection is the edge line,

$$\mathbf{p}_1 = (\hat{\mathbf{n}}, \ -\hat{\mathbf{n}}^\top \mathbf{x}_0)^\top, \quad \mathbf{p}_2 = (\widehat{\mathbf{t} \times \hat{\mathbf{n}}}, \ -\widehat{\mathbf{t} \times \hat{\mathbf{n}}}^\top \mathbf{x}_0)^\top,$$

and augment incident vertex quadrics by

$$Q_v \leftarrow Q_v + \lambda_b\,\mathbf{p}_1\mathbf{p}_1^\top + \lambda_b\,\mathbf{p}_2\mathbf{p}_2^\top,$$

with a large weight $\lambda_b$. Alternatively, restrict collapses so boundary vertices only collapse along the boundary, and forbid collapses across edges whose dihedral angle exceeds a feature threshold.

2. **Cluster construction.** We partition the simplified mesh into triangle sets $\{\mathcal{L}_k\}_{k=1}^K$ such that $\bigsqcup_k \mathcal{L}_k = \mathcal{F}$ and $\tau_{\min} \leq |\mathcal{L}_k| \leq \tau_{\max}$. For each cluster we precompute: (a) an object-space axis-aligned bounding box (AABB) $\text{AABB}_k = [\mathbf{b}_k^{\min}, \mathbf{b}_k^{\max}]$; and (b) a conservative screen-space bounding rectangle at level-0, $R_k^{(0)}$, for any given view. Project the AABB's eight corners $\{\mathbf{x}_{k,j}\}_{j=1}^8$ with the view-projection $PV$:

$$\mathbf{y}_{k,j} = PV \begin{bmatrix} \mathbf{x}_{k,j} \\ 1 \end{bmatrix}, \qquad \mathbf{u}_{k,j}^{\text{ndc}} = \left( \frac{y_{k,j}^x}{y_{k,j}^w}, \ \frac{y_{k,j}^y}{y_{k,j}^w} \right).$$

Let the viewport be $W \times H$ (origin at the top-left). Map to pixels

$$\mathbf{s}_{k,j} = \left( \tfrac{W}{2}(u_x^{\text{ndc}} + 1), \ \tfrac{H}{2}(u_y^{\text{ndc}} + 1) \right),$$

then take an outward-rounded, padded box (padding $\Delta \in \{0, 1\}$) and clip to the screen:

$$R_k^{(0)} = \left[ \lfloor \min_j \mathbf{s}_{k,j} \rfloor - \Delta, \ \lceil \max_j \mathbf{s}_{k,j} \rceil + \Delta \right] \cap [0, W{-}1] \times [0, H{-}1].$$

Such cluster construction helps us to do cluster-level culling, increasing granularity compared to per-triangle culling while retaining high selectivity.

**Per-frame visibility: frustum and Hi-Z occlusion.** Let $\{\Pi_i\}_{i=1}^6$ be the frustum planes with *inward* normals $\mathbf{n}_i$ and offsets $d_i$. A cluster $\mathcal{L}_k$ with $\text{AABB}_k$ corners $\{\mathbf{x}_j\}_{j=1}^8$ is frustum-culled if

$$\exists\, i \text{ s.t. } \max_j \left( \mathbf{n}_i^\top \mathbf{x}_j + d_i \right) < 0. \tag{10}$$

Let $Z^{(0)}(u, v)$ be the base depth. The Hi-Z pyramid for standard depth is

$$Z^{(\ell+1)}(u, v) = \max_{\delta_x, \delta_y \in \{0,1\}} Z^{(\ell)}(2u + \delta_x, 2v + \delta_y). \tag{11}$$

**Level snapping and conservative depth.** Given $R_k^{(0)}$, choose a pyramid level $\ell$ (e.g., $\ell = \mathrm{clamp}(\lfloor \log_2(\max(\mathrm{width}(R_k^{(0)}), \mathrm{height}(R_k^{(0)}))) \rfloor - c,\ 0,\ L_{\max})$ with a small constant $c \in \{1, 2\}$), and snap the rectangle to level $\ell$ by outward rounding:

$$R_k^{(\ell)} = \left[ \ \left\lfloor \frac{R_{k,\min}^{(0)}}{2^\ell} \right\rfloor,\ \left\lceil \frac{R_{k,\max}^{(0)}}{2^\ell} \right\rceil \ \right].$$

Let $\mathbf{y}_{k,j} = PV[\mathbf{x}_{k,j}^\top,\ 1]^\top$ denote the clip-space 4-vectors of the eight AABB corners introduced above (the same ones used to build $R_k^{(0)}$). A conservative near-depth estimate for the cluster is

$$\hat{z}_k \ = \ \min_{j=1,\ldots,8} \left( \max\left( z_{\mathrm{near}}^{\mathrm{ndc}},\ \frac{y_{k,j}^z}{y_{k,j}^w} \right) \right),$$

If any $y_{k,j}^w \leq 0$, the near-plane clamp above makes the estimate conservative; alternatively, one may skip the occlusion test for full safety.

Given the screen-space bounding box $R_k$ of $\mathcal{L}_k$ snapped to level $\ell$, and a conservative near depth $\hat{z}_k$ of $\mathcal{L}_k$, the occlusion test is

$$\mathrm{occluded}(\mathcal{L}_k) \iff \hat{z}_k \geq \max_{(u,v) \in R_k^{(\ell)}} Z^{(\ell)}(u, v). \tag{12}$$

**Depth-only pass with early-Z.** After visibility, we render only the surviving clusters in a *solid, depth-only* pipeline (color writes disabled, depth writes enabled). A minimal fragment shader lets the rasterizer perform early-depth testing. This produces the depth map $D \in \mathbb{R}^{H \times W}$ used downstream.

**Zero-copy interop to PyTorch.** In order to obtain the depth every frame efficiently, a naive path would be to read back the GPU depth buffer to host memory and then upload it to CUDA, introducing synchronization and PCIe traffic. Instead, we adopt a fully GPU-resident path: we render with Vulkan and export the depth image's memory as an *external file descriptor* (FD). On the CUDA side, we import that FD as external memory and map it to a device pointer; the pointer is then wrapped as a PyTorch CUDA tensor without a copy. This eliminates CPU involvement, avoids extra copies, and preserves real-time throughput.

## LLM USAGE

We acknowledge the assistance of OpenAI's ChatGPT in the preparation of this manuscript. ChatGPT was used for language refinement, LaTeX formatting, and figure illustration to improve the clarity of the presentation. All conceptual ideas, methodological designs, and final decisions were solely made by the authors.

