# OpenReview forum: "Proxy-GS: Efficient 3D Gaussian Splatting via Proxy Mesh"
_ICLR.cc/2026/Conference — ICLR 2026 Conference Withdrawn Submission_

### Official Review · Reviewer_aPkE · 2025-10-20

**Soundness:** 3
**Presentation:** 2
**Contribution:** 2
**Rating:** 4
**Confidence:** 4

**Summary:**

This paper proposes Proxy-GS, an occlusion-aware framework for MLP-based 3D Gaussian Splatting that uses a lightweight proxy mesh to (i) cull occluded anchors at test time via a fast hardware-rasterized depth map (~1k×1k under ~1 ms) and (ii) guide densification toward visible surfaces during training. The proxy-guided filter is fused with frustum culling in a single CUDA pass; proxy-guided densification projects errorful patches back to the proxy surface to grow anchors structurally. On occlusion-rich scenes (e.g., MatrixCity Streets), Proxy-GS improves quality and boosts FPS, with ablations showing best trade-offs at modest safety margins and confirming benefits from training-time consistency.

**Strengths:**

1. The paper introduces a proxy-guided Gaussian representation, which effectively leverages proxy geometry to guide Gaussian splatting for improved rendering quality and efficiency. This design elegantly integrates geometry awareness into the learning process, leading to better scene reconstruction and occlusion handling.
2. The paper is well written and clearly organized.

**Weaknesses:**

**Major**
1. The main idea of removing occluded Gaussians to accelerate rendering speed has already been proposed in OccluGaussian. The authors should cite, discuss, and compare their approach with this strong baseline.
2. The use of preprocessed proxy meshes seems quite tricky. First, I believe PGSR may not work well in very large scenes, such as the *MatrixCity small-city* dataset (which includes entire aerial or street-level environments). This is a significant limitation. Second, please provide details on the training time and storage requirements, including the time required to preprocess the proxy mesh, since users applying proxy-GS must start from scratch.
3. No demo has been submitted, so the performance of the proposed method cannot be effectively demonstrated.
4. The authors conducted experiments on only a few scenes (six in total), and the rendering metrics improve over the baselines by only 0.1 PSNR. I believe these results could easily be surpassed by tuning hyperparameters.

**Minor**
1. The paper lacks references to important related work on Level of Detail (LOD) and large-scale rendering. The authors should consider citing the following works:
    - *OccluGaussian: Occlusion-Aware Gaussian Splatting for Large Scene Reconstruction and Rendering*
    - *LODGE: Level-of-Detail Large-Scale Gaussian Splatting with Efficient Rendering*
    - *Horizon-GS: Unified 3D Gaussian Splatting for Large-Scale Aerial-to-Ground Scenes*
    - *Virtualized 3D Gaussians: Flexible Cluster-based Level-of-Detail System for Real-Time Rendering of Composed Scenes*
    - *Vast 3D Gaussians for Large Scene Reconstruction*

**Questions:**

please see the weakness above.

---

> ### Author Response · Authors · 2025-11-12
>
> 1. Thank you for pointing out OccluGaussian. We will cite and discuss this work in the revised version. A direct quantitative comparison is difficult at the moment because, to our knowledge, OccluGaussian has not released code or models, so a controlled experiment under our data and renderer is not feasible.
> Conceptually, our method differs in how occlusions are evaluated and applied. OccluGaussian partitions the scene into pre-defined regions and performs occlusion reasoning at that level, while our method performs per-pixel, proxy-guided occlusion filtering, which lets us preserve fine details and better align with the actual rendering cost. In addition, our motivation is slightly different: we mainly target MLP-based 3DGS renderers that are slower and show that proxy-guided filtering can make them practical without degrading image quality.
>
> 2. We agree that blindly assuming a high-quality mesh for very large outdoor scenes (e.g., MatrixCity-style street/aerial environments) would be a limitation. In our experiments we already included an outdoor street-scale scene (“smallcity” in the table), which is a large scene reconstructed with CityGS-X, not with the indoor PGSR pipeline. In other words, for large outdoor scenes we rely on the existing city-scale reconstruction pipeline, and the proposed proxy-GS is applied on top of that.
>
> The PGSR-style proxy was only used for smaller indoor scenes, and we additionally strengthened its depth saliency to make it more reliable in highly occluded indoor layouts. We will make this distinction clearer and move the corresponding mesh visualizations (already in the appendix) closer to the main text.
>
> Regarding preprocessing and storage: we can report in the revision (i) the one-time mesh preprocessing time per scene, (ii) the extra storage for the proxy, and (iii) the training time with and without proxy guidance, so that users who start from scratch can see the actual overhead. Importantly, this preprocessing is done once per scene and does not affect per-frame rendering speed at test time. Our motivation is to accelerate MLP-based GS rendering speed in heavily occluded scenes, where the LOD method is not enough.
>
> 3. We understand the concern. Our method is implemented in a CUDA-based renderer and already runs with the reported numbers; we will provide a demo in the revision
>
> 4. As stated in Implementation Details, all baselines and our method were trained with the same hyperparameters to ensure a fair comparison. **Since MLP-based 3DGS already has relatively high rendering efficiency, the main objective of this paper is to accelerate rendering without hurting quality**. Across these six scenes — including a virtual scene (MatrixCity), real indoor scenes (Berlin), real outdoor street scenes (SmallCity), and almost non-occluded aerial-view scenes (CUHK) — we believe the results are sufficient to demonstrate the robustness of our method.

---

> ### Comment · Reviewer_aPkE · 2025-11-13
>
> 1. I have serious concerns about the experiments in this paper. As Reviewer 2 pointed out, there is only *weak improvement over Octree-GS*, and I actually have the same question. For example, on the Berlin dataset, the PSNR is only 0.02 higher than Octree-GS, and the FPS is higher by 12. However, based on my experience, even on the same machine, running the code several times can easily lead to FPS differences within 20, which is quite normal. In my view, on the Berlin dataset, proxy-gs and Octree-GS perform essentially the same.
>    For the same reason, I would like you to evaluate more scenes to verify the effectiveness of proxy-gs. I do not think these few datasets are sufficient to demonstrate robustness. I believe you should at least add all scenes from Mega-NeRF and UrbanScene3D, the Campus dataset from Hierarchical3DGS, and the Alameda, London, and NYC datasets from ZipNeRF, and make direct comparisons with the metrics reported in the LODGS paper. At the same time, I would like to know why, compared with the results in the LODGE paper, the metrics for SmallCity are much worse.
>
> 2. As the first reviewer also asked: how much time does it take to extract the proxy meshes, and how much storage do they require? I need you to provide these numbers to me, rather than only stating them in the revision.
>
> 3. I would like to know why you do not conduct experiments on the entire Small City in MatrixCity, but instead split it into several blocks. Is this because it is not possible to obtain a good proxy mesh for the whole area?
>
> 4. I still feel that the assumptions behind proxy-mesh are somewhat too strong, and it is not entirely fair to compare with the baseline after adding this component. Once you have such a proxy mesh, you could also use it, like GSDF, to exploit the mesh and further improve the rendering quality of GS.

---

> ### Author Response · Authors · 2025-11-13
>
> Thank you for your reply.
> 1. On the Berlin dataset. Please refer to Table 9 in the appendix. Although Berlin does contain occlusions, the overall scene scale is small. When the number of anchors is below 100k, anchor decoding runs at nearly the same speed across methods. The differences emerge only when scaling to larger indoor environments (unfortunately, we do not have a public dataset at that scale). From an application standpoint, for small scenes like Berlin, existing methods are already fast enough; our goal is to address the bottlenecks that appear at scale. We have already discussed in the paper that for top-down scenes, an LOD-style design is generally sufficient. Both Mega-NeRF and UrbanScene3D fall into this aerial-view category. Our motivation, however, is to address heavily occluded scenes, where LOD alone cannot resolve the occlusion-related inefficiencies. As discussed with CUHK-LOWER, top-down scenes contain very limited occlusion, and we explicitly analyze this behavior in the paper. For the Campus scene, we plan to include additional results in the revised version.
>
> Regarding your suggestion of comparing directly with LODGE, for Alameda-style scenes, you can refer to our Berlin results, since they are of the same type and scale. These scenes are not genuinely large.  Our motivation, however, is to address the LOD method's bottleneck in large occluded scenes（Matrixcity street). As for the lower performance on Small City, we believe this mainly comes from the training schedule. If you are familiar with the Hierarchical-3DGS dataset, Small City contains over 5,000 images, and Campus even more. However, the LODGE paper states in Datasets & Baselines that each scene contains only 1,000–2,000 images. Therefore, although they report 40k training steps with an octree, it is likely trained on pre-split chunks by hierarchical-3dgs, while we trained the entire scene directly. This difference explains why the reported Octree-GS numbers can diverge.
>
>
>
> 2. Preprocessing time and storage. The required time depends on the data modality and reconstruction route. If dense point clouds or LiDAR are available, the proxy can be obtained directly with virtually no extra preprocessing (<10min). For RGB-only cases, using the full MatrixCity scene (~5,000 images) as an example, proxy generation takes about 5 hours (measured end-to-end on our workstation; hardware details provided in the supplement). The storage overhead is small thanks to our poly reduce: in the Block-E scene, we compress the original mesh from 108 MB to 874 kB with negligible loss in rendering quality.
> Since this point drew considerable attention, we ran additional ablation experiments on Block E last night.
> | Mesh Size | 108.9 MB | 7.9 MB | 4.7 MB | 874 KB |
> | --------- | -------: | -----: | -----: | -----: |
> | PSNR (dB) |    21.67 |  21.72 |  21.66 |  21.68 |
>
> 3. Why split Small City into blocks instead of training on the entire area?
> This choice is not due to difficulties obtaining a proxy mesh for the whole area. In city-scale reconstruction, it is common to reconstruct per-block and then fuse (e.g., CityGaussianV2, ULSR-GS), so producing a single global mesh or a block-wise mesh is algorithmically equivalent for our pipeline. We split Small City primarily for efficiency and stability: the street-scale scene contains 8k+ images, and block-wise training converges faster, fits GPU memory more reliably, and allows parallelization without changing the method or hyperparameters. Conceptually, if we render using all blocks (i.e., the full area), the speedup of our method over the baseline would be even greater.
>
> 4. We understand the concern. Our paper’s motivation is acceleration under heavy occlusion, not quality enhancement. **Per-pixel occlusion culling in GS is known to be either fragile or time-consuming**; our contribution is to make this culling robust and cheap by using a lightweight proxy purely as a visibility prior. The proxy is not used to improve shading or geometry fidelity; it only gates which Gaussians should be evaluated.
> Regarding fairness: one could also combine a mesh with other GS variants (e.g., GSDF) to improve quality, but that does not address the render-time cost, which is our target. We actually tested “just add a mesh” to a strong baseline (Octree-GS); see Table 4, ID2—it does not yield the speed/quality behavior we seek. What matters is the culling pipeline and its alignment with rendering, not merely having a mesh. Our training aligns GS to the proxy’s occlusion structure; skipping this alignment leads to a distribution mismatch and degraded results (again reflected by Table 4, ID2).

---

> > ### Author Response · Authors · 2025-11-13
> >
> > Thank you for the constructive and responsible suggestions. Due to the rebuttal process, we cannot include additional visualizations at this stage, but we will include them comprehensively in the feature version and public release, including qualitative and quantitative results using a proxy with different noise and resolution. If there is no additional feedback from other reviewers by tomorrow, we plan to withdraw the paper by tomorrow.

---

### Official Review · Reviewer_b4Nz · 2025-10-23

**Soundness:** 2
**Presentation:** 3
**Contribution:** 2
**Rating:** 2
**Confidence:** 4

**Summary:**

This paper proposes Proxy-GS, a framework for accelerating MLP-based 3D Gaussian Splatting (3DGS) by introducing a proxy-mesh–guided occlusion filter and densification mechanism. The core idea is to pre-compute a lightweight proxy mesh to perform depth-based occlusion culling via hardware rasterization (<1 ms) and use these depth maps both during training (to guide anchor densification) and inference (to cull occluded anchors). The authors claim that this approach improves efficiency and rendering quality simultaneously, reporting up to a 3× FPS increase and slight PSNR gains over Octree-GS and other baselines on datasets such as MatrixCity, Small City, and CUHK-LOWER.

**Strengths:**

1. The paper addresses a relevant efficiency problem in MLP-based 3DGS rendering, which is indeed computationally heavy.

2. The proposed integration of proxy meshes into training/inference is intuitive and could be useful for applied rendering systems.

**Weaknesses:**

1. No theoretical grounding: The proxy-guided densification lacks a clear optimization objective or loss formulation beyond heuristics; equations (2–9) only restate camera transformations.

2. Weak improvement over Octree-GS: The reported gains compared with Octree-GS are modest and often within the margin of experimental noise. Moreover, since the entire codebase and training pipeline are directly built upon Octree-GS, it is difficult to disentangle whether the improvements truly originate from the proposed proxy mechanism or from tuning and implementation differences. This raises doubts about the actual effectiveness and generality of the method.

**Questions:**

1. How robust is Proxy-GS to inaccurate or incomplete proxy meshes? Quantitative sensitivity analysis is missing.

2. Why is “occlusion-guided densification” superior to simply pruning via hierarchical Z-buffering or view-space LOD strategies?

3. Were all methods trained with identical data augmentations, learning rates, and iteration counts?

---

> ### Author Response · Authors · 2025-11-12
>
> 1. densification lacks clear optimization objectiveness beyond heuristics?
> Several methods [3DGS-MCMC, mini-splatting] have verified that densifying Gaussians at certain areas indicated by rendering error maps can help the convergence of training and improving training quality.  Instead of using 2D rendered error maps, our surface-oriented densification directly takes the insight that Gaussians should appear at the correct depth, and we thus generate new Gaussians at these locations. As supported in the ablation study Table. 4, the proposed densification improved the PSNR due to better structured anchors organizations.
> 2. equations (2–9) only restate camera transformations?
> Equations (2)-(5) are background knowledge of how the Gaussians are projected to get the depth value, and they were originally used in Gaussian rendering but not in the Gaussian filtering. We add them into Gaussian filtering and check the z value with respect to mesh-rendered z value.
> Additionally, we check the projected position by Equation (6) to filter out of image region Gaussians. Then we transform the hardware-rasterized depth into comparable depth by Equation (7) and (8), and filter Gaussians whose projected depth is farther than the transformed hardware-rasterized depth by Equation (9).
> As a summary, only (2)-(5) are seen in Gaussian Splatting, and they were originally not purposed for filtering, we add them to filtering. And we also add additional constrains (6)-(9), where the hardware-rasterized depth is also our contribution.
> Weak improvement over Octree-GS?
> 3. Weak improvement over Octree-GS?
> As seen from Table. 1, we accelerate the rendering speed of Octree-GS by 3.14x to 4.47x while maintaining the rendering quality (actually we even slightly improve the rendering quality due to our proxy-guided densification, whose effect is ablated in Table 4.)
> 4. Do the improvements originate from our proposed method or just tuning hyperparameters?
> We do NOT accept rude and baseless doubt about our academic ethics. We have explicitly stated in Section 5: Implementation Details that all baselines and our method have used the same hyperparameters, including learning rate, training iterations, as well as other Octree-GS’s hyperparameters. The PSNR improvements come from our Occlusion-Aware Training and Proxy-Guided Densification, as explicitly ablated in Table. 4.
> 5. How robust is our method to inaccurate or incomplete meshes?
> We will illustrate this more with additional experiments.

---

### Official Review · Reviewer_P1v1 · 2025-10-28

**Soundness:** 2
**Presentation:** 3
**Contribution:** 2
**Rating:** 6
**Confidence:** 4

**Summary:**

This paper proposes Proxy-GS, an occlusion-aware and proxy-guided training and rendering framework for 3D Gaussian Splatting (3DGS). The central idea is to introduce a lightweight proxy mesh derived from SfM/MVS pipelines, and to use it as a structural scaffold to: 1) Guide densification toward geometrically meaningful regions. 2) Enable occlusion-aware pruning and rendering. 3) Reduce redundancy while maintaining reconstruction quality.

**Strengths:**

Introducing proxy-guided Gaussian growth is conceptually elegant and practically useful.

Results are good, particularly in occlusion-heavy environments.

Visualizations effectively reveal how proxies shape spatial distribution.

**Weaknesses:**

1. Dependence on Proxy Quality:

If the proxy mesh is coarse / noisy, how robust is Gaussian distribution?

Ablations on proxy resolution and noise are missing.

2. Fairness of Experimental Comparison:

Some baselines are not clearly described regarding training iterations, learning rate tuning, and hardware.

Include a wall-clock cost per iteration comparison.

3. Scalability Beyond Desktop-Scale Scenes:

Claims target “ultra-large VR walkthroughs” but datasets are still mid-scale.

Evaluate on a truly large-scale multi-block urban dataset (e.g., Tanks&Temples extended / UrbanScene3D).

4. LOD vs. Occlusion Interactions:

The method implicitly resembles geometric LOD.

Discuss when proxy-GS degenerates to simple LOD behavior.

**Questions:**

1. What failure behaviors emerge when the proxy mesh is inaccurate or incomplete?

2. How sensitive is the densification policy to mesh vertex density and normal consistency?

3. Is the occlusion culling performed per-ray or per-fragment? Clarify computational impact.

4. Could proxy features replace Gaussians in distant regions entirely, forming a hybrid representation?

---

> ### Author Response · Authors · 2025-11-13
>
> 1. Thank you very much for the thoughtful suggestions. We have completed ablations of the proxy under two stressors: (i) mesh resolution (fine→coarse) and (ii) vertex noise. Vertex noise denotes the world-space magnitude of random perturbations applied to mesh vertices. Results are as follows.
>
> (A) Proxy resolution (mesh size) vs. PSNR
> | Mesh Size | 108.9 MB | 7.9 MB | 4.7 MB | 874 KB |
> | --------- | -------: | -----: | -----: | -----: |
> | PSNR (dB) | 21.67 | 21.72 | 21.66 | 21.68 |
>
> (B) Vertex noise (world-space magnitude) vs. PSNR
> | Vertex noise percent | 0% | 5% | 10% | 20% |
> | --------- | -------: | -----: | -----: | -----: |
> | PSNR (dB) | 21.67 | 21.61 | 19.11 | 16.74 |
>
> These results demonstrate that our method retains a certain degree of robustness to perturbations. Due to the rebuttal policy, we are not able to provide mesh visualizations at this stage. We will add the visualization in the future revision.
>
> 2. Fairness of Experimental Comparison. As detailed in Implementation Details, all methods are trained under the same settings (including 40k training steps); we do not tune baselines differently from our method.
>
> 3. Tanks&Temples is quite small, and UrbanScene3D is a larger top-down (aerial) dataset similar to CUHK-LOWER; because top-down imagery has minimal occlusion, We also analyze the expected gains in these top-down scenes (CUHK-LOWER).
>
> 4. proxy-GS degenerates to simple LOD behavior. You can refer to our ablation study Table 4.

---

### Official Review · Reviewer_1Spc · 2025-10-30

**Soundness:** 2
**Presentation:** 2
**Contribution:** 2
**Rating:** 2
**Confidence:** 5

**Summary:**

This work introduces Proxy-GS, a large-scale 3D Gaussian Splatting (3DGS) framework designed for both rendering and training. The proposed method utilizes a proxy mesh for occlusion culling, which facilitates both proxy-guided rendering and densification. Experimental results on multiple datasets demonstrate the effectiveness of the approach.

**Strengths:**

This work on large-scale real-time scene rendering and training represents a timely and practical direction for 3D Gaussian Splatting.

**Weaknesses:**

Introduction
1. Line 096. One of the main contributions 'We leverage engineering optimizations ...' seems to describe an implementation rather than a methodological contribution.



METHOD
1. The framework is not clearly illustrated, and Figure 2 is not referenced anywhere in the manuscript.
2. The proposed method heavily relies on a pre-generated mesh as a proxy, which limits its practicality. Moreover, the use of such a proxy may result in an unfair comparison with mesh-free approaches.
3. The motivation for the proxy-guided filter is unclear. It is feasible to extract visibility or depth for occlusion estimation directly from 3DGS models. Why is an additional mesh necessary?
4. Equations (2) to (8) appear to replicate the culling implementation of the original 3DGS method, but the manuscript does not provide appropriate citations.


EXPERIMENT
1. It would be helpful to provide the computational or time budget required for obtaining the mesh.
2. It would be useful to include failure cases resulting from unsuccessful mesh reconstruction. For example, surface reconstruction may fail to recover fine details in large-scale scans.

Typo:

line 230: interpretability -> interpretability.

**Questions:**

1. This manuscript is not well-written, and it is recommended to restructure the manuscript.
2. It would be better clarify the necessity of proxy mesh.

---

> ### Author Response · Authors · 2025-11-12
>
> We would like to clarify that the goal of this work is to **accelerate rendering in heavily occluded scenes**. Our intention is not to assume a strong, high-quality mesh as a prerequisite. In fact, we tested directly plugging the mesh into Octree-GS, and the result is reported in Table 4, ID2: the performance is not satisfactory, which shows that “using a mesh” alone is not what brings the improvement.
>
> The **motivation** is already clearly stated in the first two sentences of Sec. 4.2: we need a fast, view-consistent way to estimate occlusion so that we can safely filter Gaussians without hurting image quality and accelerate rendering speed. In principle, one could extract visibility or depth directly from 3DGS, but most alternative depth/visibility estimators are either (i) more time-consuming, or (ii) not multi-view consistent, which contradicts our goal of “faster without sacrificing quality.” The proxy gives us a cheap and stable geometric reference to guide filtering, so we do not think the motivation is problematic.  For example, you mentioned method "extract visibility or depth for occlusion estimation directly from 3DGS models" will double the rendering time.
>
> Please look closely at the formulas. Equations (2)–(5) correspond to the standard 3DGS-style frustum culling procedure, and we will add explicit citations to the original 3DGS paper to make this clear. Equations (5)–(8), however, are newly introduced for our CUDA-based renderer, mainly to preserve fine details under our proxy-guided filtering scheme. So this part is not a simple copy of existing work but an implementation extension specific to our method.
> Moreover, one of our key contributions is that, by introducing a lightweight and explicit occlusion prior (a proxy mesh), we are able to tightly couple hardware rasterization with 3DGS CUDA-based rendering. In the contribution section, we have also highlighted that this design reduces the depth rendering cost to unprecedented 1 ms. Since this part involves a number of formulas and implementation details, we placed the full derivation in the appendix and only present the more intuitive components in the main text.

---

### Note · Authors · 2025-11-14

**Comment:**

We feel that this submission may not be fully aligned with the reviewers’ primary areas of expertise, which has resulted in several misunderstandings about our work. In light of this, we have decided to withdraw the paper.

**Withdrawal Confirmation:**

I have read and agree with the venue's withdrawal policy on behalf of myself and my co-authors.